# Compendium of medically important snakes, venom activity and clinical presentations in Ghana

Justus Precious Deikumah[1], Robert Peter Biney[2], John Koku Awoonor-Williams[3], Mawuli Kotope Gyakobo [4]*

1 Department of Conservation Biology and Entomology, School of Biological Sciences, College of Agriculture and Natural Sciences, University of Cape Coast, Cape Coast, Ghana, 2 Department of Pharmacotherapeutics and Pharmacy Practice, School of Pharmacy and Pharmaceutical Sciences, College of Health and Allied Sciences, University of Cape Coast, Cape Coast, Ghana, 3 Independent Health System and Policy Analyst, Cantonments, Accra, Ghana, 4 Department of Internal Medicine and Therapeutics, School of Medical Sciences, College of Health and Allied Sciences, University of Cape Coast, Cape Coast, Ghana

* mawuli.gyakobo@ucc.edu.gh

**Data Availability Statement:** The authors confirm that all data underlying the findings are fully available without restriction. All relevant data are

## Abstract

### Background

Snake bite envenoming (SBE) is one neglected tropical disease that has not received the needed attention. The sequelae of burdensome disability and mortality impact the socioeconomic life of communities adversely with little documentation of SBE in health facility records in Ghana. This study details SBE and snake distribution, habits/habitats, type of venom expressed and clinical manifestations.

### Methodology

We conducted a structured thematic desk review of peer reviewed papers, books and reports from repositories including PubMed, World Health Organization (WHO) and Women's & Children's Hospital (WCH) Clinical Toxinology Resources using bibliographic software EndNote and search engine Google Scholar with the following key words; snakes, medical importance, snake bites, venom and venom type, envenomation, symptoms and signs, vaccines, venom expenditure, strike behaviour and venom-metering + Ghana, West Africa, Africa, World. We also reviewed data from the District Health Information Management System (DHIMS) of the Ghana Health Service (GHS). Outcome variables were organized as follows: common name (s), species, habitat/habit, species-specific toxin, clinical manifestation, antivenom availability, WHO category.

### Findings

Snake bites and SBE were grouped by the activity of the expressed venom into neurotoxic, cardiotoxic, haemorrhagic, cytotoxic, myotoxic, nephrotoxic and procoagulants. Neurotoxic snake bites were largely due to elapids. Expressed venoms with cardiotoxic, haemorrhagic, nephrotoxic and procoagulant activities principally belonged to the family Viperidae. Snakes

within the paper and its Supporting Information files.

**Funding:** The author(s) received no specific funding for this work.

**Competing interests:** The authors have declared that no competing interests exist.

with venoms showing myotoxic activity were largely alien to Ghana and the West African sub-region. Venoms showing cytotoxic activity are expressed by a wide range of snakes though more prevalent among the Viperidae family. Snakes with neurotoxic and haemorrhagic venom activities are prevalent across all the agro-ecological zones in Ghana.

## Conclusion/Significance

Understanding the characteristics of snakes and their venoms is useful in the management of SBE. The distribution of snakes by their expressed venoms across the agro-ecological zones is also instructive to species identification and diagnosis of SBE.

### Author summary

We observed rudimentary data on snake bites and a paucity of venom characterization in the health records of patients in health facilities in Ghana. Knowledge of the distribution of snake species across the agro-ecological zones of the country was also limited. We set out to complement this data by doing a structured thematic desk review of peer reviewed papers, books and reports from repositories including PubMed, WHO and the District Health Information Management System (DHIMS) of the Ghana Health Service (GHS).

Our findings revealed that, snake bites were largely grouped according to the activity of the expressed venom into neurotoxic, cardiotoxic, haemorrhagic, cytotoxic, myotoxic and in some documentations nephrotoxic and procoagulants. Neurotoxic snake bites were largely due to cobras and mambas. Snake expressing venoms with cardiotoxic, haemorrhagic, nephrotoxic and procoagulant activities were principally vipers. Snakes with venoms showing myotoxic activity were largely alien to Ghana and the West African sub-region. Venoms showing cytotoxic activity are expressed by a wide range of snakes though more prevalent among vipers.

In Ghana, snakes expressing venoms with neurotoxic and haemorrhagic activities are prevalent across all the agro-ecological zones of the country. We identified 3 antivenoms registered by the Ghana Food and Drug Authority.

## 1 Introduction

Snakes are seen as cryptograms of fear and discomfort to human societies throughout the ages due to the danger associated with their bites as well as misconceptions about their biology and behaviour. Global annual reports of snake bites exceed 5 million according to WHO [1]. While a respectable number of snake bites are not venomous [2] envenoming from venomous bites accounts for more than one-third of this global incidence resulting in significant morbidity including amputations and permanent disabilities and death [1].

Snake bites remain a neglected emergency in tropical Africa due to low awareness of snake bites as a public health problem in many of these countries [3,4]. An estimated 435,000 to 580,000 snake bites occur in Africa annually that need treatment [1]. Snake envenoming affects women, children and farmers in poor rural communities in low- and middle-income countries but the highest burden occurs in countries where health systems are weakest and medical resources rudimentary [1]. Hospital-based data from only sub-Saharan Africa puts incidence at 56.4/100,000 and mortality at 1.35/100,000 inhabitants in rural areas [5,6]. In most instances, this data is only a microcosm of actual snake bite data as the reported data is largely

from a few parts in Africa noted for notoriously high rates of snake bites [7]. For example an estimation in northern Ghana reported snakebite incidence of 56.4/100,000 and mortality at 1.35/100,000 inhabitants [8].

Management of venomous snake bites require timely use of appropriate pharmacological interventions including the use of antivenoms to minimize the progression of tissue damage and systemic toxicity. Although efficacious antivenom exists, the capacity to produce is a challenge for most regions where snake bites and snake envenomation is endemic while existing manufacturers of snake antivenom are gradually scaling down [9]. Compounding these challenges is the proliferation of poor quality antivenoms available in these areas. This coupled with the fact that there should be separate antivenoms for different families of snakes makes the situation dire and gravely affect access to treatment [9,10].

In Northern Ghana, there were an estimated 86 envenomings and 24 deaths/100,000/year caused mainly by *Echis ocellatus* [5] while other studies in the Brong-Ahafo Region of Ghana found snakebite incidence of 92/100,000 [11]. Emerging from these studies is that, complete epidemiological data on snakebites are inadequate suggesting that most cases are never reported. This may be due to lack of confidence in the medical system for inadequately treating snake bite envenoming (SBE), poorly resourced health facilities in handling SBE or dearth in competence among health workers in recognizing and characterizing SBE and affording appropriate treatment. The myths and traditional belief systems in some communities may as well have exacerbated the challenges associated with this public health emergency.

In Ghana, the electronic repository, District Health Information Management System (DHIMS) of the Ghana Health Service captures snakebites data without envenomation characteristics and venom specific diagnoses probably because this data is lacking in health care facilities due to inadequate knowledge in the epidemiology of snakes, snake bites and the types of envenomation with their clinical manifestation. This impacts on the quality of care for snake bite victims. Understanding the clinical manifestation of SBE and associated snake type could reduce the unnecessary use of polyvalent anti snake sera with its concomitant side effects.

This study seeks to fill this gap with a structured thematic desk review that characterizes medically important snakes by name, ecology, habitat, distribution, associated venom type expressed, clinical presentations and snake antivenom availability in Ghana and the West African sub-region. This knowledge when available, will improve diagnosis of snake bites and ultimately enhance treatment of snake bite envenomation in Ghana and the west Africa sub-region. Information provided in this paper will significantly improve the understanding of snakes and their behaviour, snakebite types and the management of snakebites as well as the quality of data captured into the electronic database of Ghana's District Health Information Management System (DHIMS).

## 2 Methods

### 2.1. Study area

The Republic of Ghana, is a country in West Africa (7.9465° N, 1.0232° W) that spans the Gulf of Guinea and the Atlantic Ocean to the south, and shares borders with the Ivory Coast in the west, Burkina Faso in the north, and Togo in the east (Fig 1). Ghana covers an area of 238,535 km$^2$ (92,099 sq mile), spanning diverse biomes that range from coastal savannas to tropical rainforests. With a population of 30.8 million people, Ghana is the second-most populous country in West Africa, after Nigeria. Over 43% of Ghana's population resides in the rural parts of the country with slightly more males (51%) than females (49%) [12]. The main economic activity of the rural poor inhabitants is subsistence agriculture while over 11% of the total population of the country earn less than 2 USD per day which is below the poverty line [13].

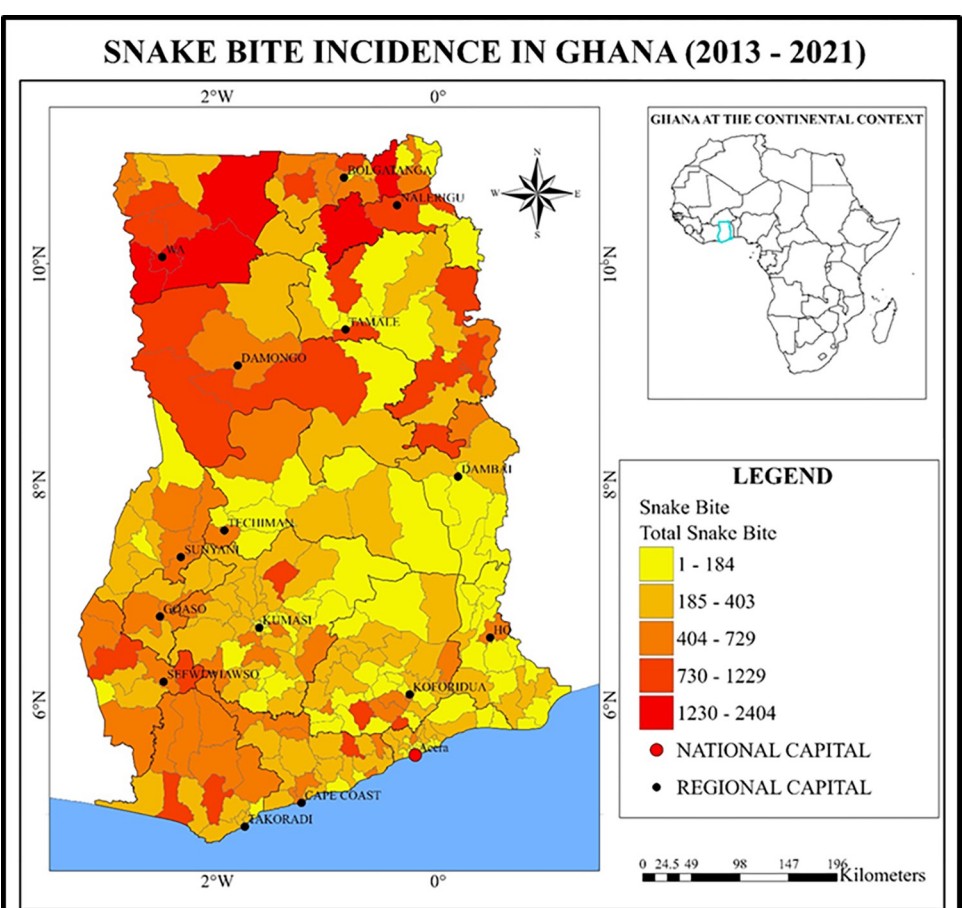

**Fig 1. Snakebite incidence distribution hotspot Map of Ghana.** Data source: Demographic Health Information Management System (DHIMS) of the Ghana Health Service (GHS)– 2013–2021.

The climate of Ghana is tropical with a warm and dry eastern coastal belt, a hot and humid south-west corner and hot and dry northern belt [14]. There are two main seasons: wet and dry. With the global climate change, the north of Ghana experiences the rainy season from April to mid-October while in the south its rainy season is from March to mid-November. The harmattan, a dry desert wind, blows in north-east Ghana from December to March, lowering the humidity and causing hotter days and cooler nights in the northern part of Ghana. Average daily temperatures range from 30°C (86°F) during the day to 24°C (75°F) at night with a relative humidity between 77% and 85%. In the southern part of Ghana, there is a bi-modal rainy season: April through June and September through November. Storms occur in the northern part of Ghana during March and April, followed by occasional rain until August and September, when the rainfall reaches its peak. Rainfall ranges from 78 to 216 cm (31 to 85 inches) a year. These variations in weather and climatic parameters determine agricultural activities in the five agro-ecological zones.

## 2.2. Data search strategy

Information for this work was mainly gathered from a structured thematic desk review of secondary data. Peer reviewed papers, books and other reference sources were collated from repositories including PubMed, WHO and Women's & Children's Hospital (WCH) Clinical

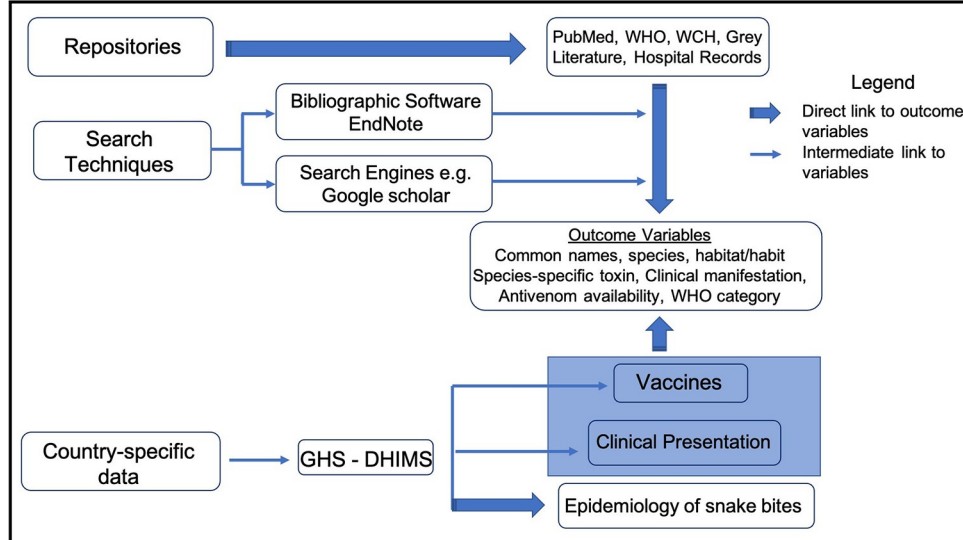

**Fig 2. Flow chart showing information gathering process.**

Toxinology Resources using bibliographic software EndNote and search engine Google Scholar among others with the following key words; snakes, medical importance, snake bites, venom and venom type, envenomation, symptoms and signs, vaccines, venom expenditure, strike behaviour and venom-metering + Ghana, West Africa, Africa, World (Fig 2). Additionally, data from the Ghana Health Service (GHS)'s District Health Information Management System (DHIMS) spanning from 2013 to 2021 was reviewed for snake bites (Fig 2).

Information gathered from these reference sources were grouped into the following outcome variables: Common name (s), Species, Habitat/habit, Species-specific toxin, Clinical manifestation, Antivenom availability, WHO category (Fig 2).

## Data management

The information collected was displayed in a matrix according to the following themes: Common name (s), Species, Habitat/habit, Species-specific toxin, Clinical manifestation, Antivenom availability and WHO category. Information was retrieved from multiple sources for comparison and validation along the themes and outcome variables. The matrix was cleaned of irrelevant information or unproven assertions. Final validated information was then grouped in tables along the outcome variables according to the type of venom expressed including, neurotoxin, haemorrhagin, cardiotoxin, myotoxin and cytotoxin. The distribution and type of snakes are displayed in agro-ecological maps of Ghana. Snake types are symbolically displayed in these ecological zones across the country (Fig 3).

## 3 Results

### 3.1: Characterisation of medically important snakes in Ghana and the sub-Region by name, ecology, habit and distribution patterns

**3.1.1: African snakes and characteristics of expressed venoms.** Snakes of medical importance are usually grouped according to the activity of their expressed venoms. The following groups have been identified, neurotoxic, myotoxic, cardiotoxic, haemorrhagic, procoagulant and cytotoxic (Table 1). Snakes expressing venom with procoagulant activity are

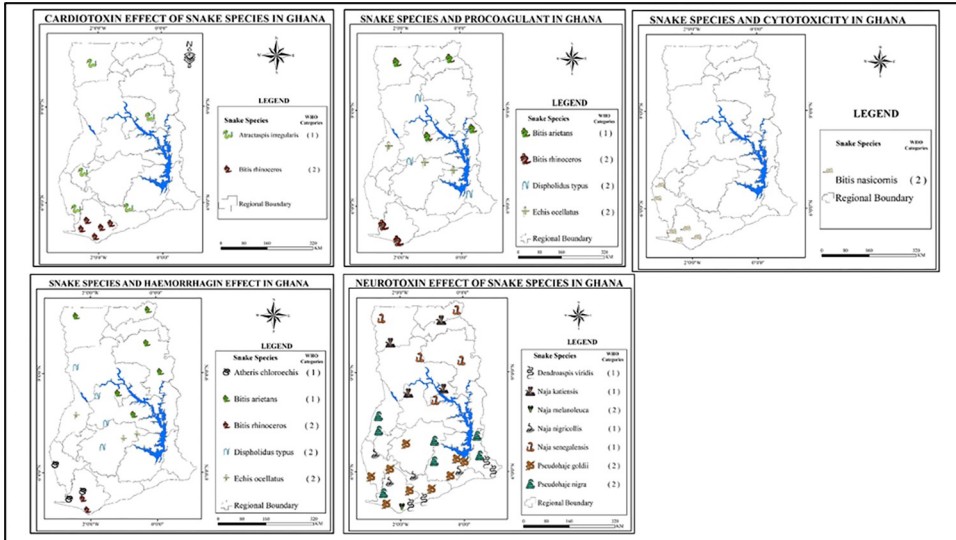

**Fig 3. Characterisation of medically important snakes in Ghana by name, ecology, habit and distribution patterns.** Data sourced from grey literature and additionally data adapted from Global biodiversity information facility (https://www.gbif.org/). Maps are drawn using shapefiles from free sources (U.S. Geological Survey (USGS) (http://www.usgs.gov)) while icons are sourced from commons.wikimedia.org.

sometimes grouped with the haemorrhagic set. Some literature also has nephrotoxic group that clinically manifest as acute kidney injury and are quite often caused by snakes expressing venoms with haemorrhagic activity mostly from the Viperidae family.

The distribution of these medically important snakes by the activity of their expressed venoms across the agro-ecological zones in Ghana is demonstrated in Fig 3. Snakes expressing venoms with haemorrhagic and neurotoxic activities were found to be the most prevalent.

**Table 1. African snakes and characteristics of expressed venoms [2,15–22].**

| Venom activity | Clinical manifestation | African snake responsible |
|---|---|---|
| Neurotoxicity | Local symptoms: parasthesias, neuropathic pain<br>Systemic symptoms: Ptosis, external ophthalmoplegia, facial paralysis, paralysis of tongue, inability to open mouth, bulbar and respiratory paralysis<br>Venom ophthalmia: pain, hyperemia, blepharitis, blepharospasm and corneal erosions | Elapids (e.g. Cobras, Rinkals, Mambas); Berg adder and Peringuey's adder, selected Bothrops spp. |
| Myotoxicity | Local symptoms: Swelling /Oedema, subcutaneous bleed<br>Systemic symptoms: Trismus, stiff painful muscles, myoglobinuria, rhabdomyolysis /muscle necrosis | Sea snake, European adder (*Vipera berus*), some Viperidae snake bites (selected Bothrops spp.) |
| Cardiotoxicity | Hypotension, cardiogenic shock, arrhythmias with sinus node dysfunction, ECG abnormalities, raised serum/cardiac enzymes with acute myocardial infarction and/or ischaemia, myocarditis and acute pulmonary oedema | Puff Adder and other giant *Bitis* species, saw scaled vipers (*Echis ocellatus*) |
| Haemorrhagic | Local symptoms: painful progressive swelling and tissue destruction<br>Systemic symptoms: Spontaneous systemic haemorrhage (gums, gut, brain, etc) associated with venom-induced consumption coagulopathy | Saw-scaled vipers (*Echis ocellatus*), puff adder and other large Bitis species, boomslang, Forest vine snake (*Thelotornis kirtlandii*) |
| Procoagulant | Coagulopathy: incoagulable blood with haematemesis and persistent bleeding from trauma sites and recent wounds | Saw-scaled vipers, boomslang (*Dispholidus typus*), Forest vine snake (*Thelotornis kirtlandii*) |
| Cytotoxicity | Massive local swelling, blistering, necrosis: extravasation of blood and plasma potentially causing hypovolaemia | Puff adder and other large Bitis species, saw-scaled vipers, spitting cobras |

## 3.2. Characterisation of medically important snakes of Ghana and the West African sub-region, associated expressed venom type and activity with clinical presentations (Tables 2–6)

**3.2.1. Snake species with neurotoxic effects in humans.** Generally, snake species in Ghana with neurotoxicity effects are cobras and mambas. We found seven different cobras and 1 green mamba that have neurotoxins. The seven cobra species are from the same family, four of which are from the genus *Naja* and the other 2 from genus *Pseudohaje*. The green mamba species is the Western Green Mamba also known as the Hallowell's Green Mamba (*Dendroaspis viridis*.) The snakes with neurotoxins are found in a variety of habitats that are generally distributed in all agro-ecological zones in Ghana with many of them inhabiting areas near water and dry land (Fig 3). A few are arboreal and nocturnal (Table 2). Most of the toxins expressed in these snakes act post-synaptically to produce clinical symptoms including mild to moderate flaccid paralysis with severe blistering. Out of the seven snakes in this category, there are various specific antivenoms available for all except the Gold's Tree Cobra (*Pseudohaje goldii*) and Black Tree Cobra (*Pseudohaje nigra*). There is indication that at present the specific neurotoxin in these *Pseudohaje species* remain unknown hence no specific antivenoms available (Table 2). We found that, out of the seven species with neurotoxic effects, 3 of them are listed as WHO category 2 medically important highly venomous snakes capable of causing significant morbidity and death for which exact epidemiological and clinical data is lacking. The remaining 4 belong to Category 1 highest medically important venomous snakes that are

**Table 2. Snakes species with neurotoxic (Postsynaptic) effects in humans [23, 24].**

| Species | Habitat/habit | Other toxins | Clinical manifestation | Antivenom availability |
|---|---|---|---|---|
| Forest Cobra (*Naja melanoleuca*)** | Forest, around inhabited areas and near water | Cardiotoxins | Ptosis, cranial nerve palsies, bulbar and respiratory paralysis (moderate to severe flaccid paralysis). Local effects: pain, severe swelling, bruising, blistering, necrosis | FAV-Afrique, SAIMR Polyvalent Antivenom, Antivipmyn Africa, Bivalent Naja / Walterinnesia Snake Antivenom, SAIMR Snakebite Kit |
| Black-necked spitting cobra (*Naja nigricollis*)* | Moist savanna and Nocturnal | Necrotoxins+, Cardiotoxins | Flaccid paralysis, severe swelling, bruising, blistering, necrosis, venom spit ophthalmia: kerato-conjuctivitis, corneal ulceration | Bivalent Naja / Walterinnesia Snake Antivenom, Antivipmyn Africa, Pan-African antivenom (EchiTAb-Plus-ICP), Polyvalent Snake Venom Antiserum,FAV-Afrique, Favirept |
| West African brown spitting cobra (*Naja katiensis*)* | Savanna, Diurnal, & arboreal | Cardiotoxins | Moderate to severe flaccid paralysis Severe swelling, bruising, blistering and necrosis Venom spit ophthalmia: kerato-conjuctivitis, corneal ulceration | Bivalent Naja / Walterinnesia Snake Antivenom, Antivipmyn Africa |
| Senegalese cobra (*Naja senegalensis*)* | Savanna, riparian forests | Myotoxins+ Cardiotoxins Necrotoxins+ | Moderate to severe flaccid paralysis Severe swelling, bruising, blistering and necrosis | FAV-Afrique |
| Gold's tree cobra (*Pseudohaje goldii*)** | Forest, woodland, near watercourses, Arboreal & diurnal | unknown | Insufficient data | None presently |
| Black Tree Cobra (*Pseudohaje nigra*)** | Forest, woodland, near watercourses, Arboreal & diurnal | unknown | Insufficient data | None presently |
| Western Green Mamba (*Dendroaspis viridis*)* | Forest, thicket and woodland, farms, Diurnal, arboreal | Dendrotoxins Fasciculins | Flaccid paralysis, neuroexcitation, increased sweating, salivation | FAV-Afrique, Antivipmyn Africa |

[+ = present but not defined

* = WHO category 1—Highest medically important venomous snakes

** = WHO category 2—Secondary medically important highly venomous snakes]

widespread and cause numerous snakebites, resulting in highest level of morbidity, disability or mortality. This result suggests that there is antivenom available for WHO Cat 1 species in this group but not the Cat 2 of the *Pseudohaje species* although their envenomation is highly lethal.

**3.2.2. Snake species with cardiotoxic effects in humans.** Two principal species were found to possess cardiotoxins in our study, the Variable Burrowing Asp (*Atractaspis irregularis)* and West African Gaboon Viper (Bitis rhinoceros). Both snake species are common in the forest terrain. When they occur in savanna regions, they are localized to patches of the savanna which are close to the forest regions (Table 3). A cardiotoxin, sarafotoxin, with endothelin-like properties has been isolated from the Variable Burrowing Asp but cardiotoxin associated with West African Gaboon Viper is not yet characterized. However, we found several antivenom products available for this snake species but none available for the Variable Burrowing Asp (Table 3). Bites from these species can cause coagulopathy, infarction, and ischemia.

**3.2.3. Snake species with haemorrhagic/haematotoxic effects in humans.** With regards to snake species that have toxins that are haemorrhagins, we found 5 different species of snakes of which 3 are vipers i.e. the West African Carpet Viper (*Echis ocellatus*), West African Green Tree Viper, (*Atheris chloroechis*) and West African Gaboon Viper (*Bitis rhinoceros*). These three viper species are mostly found in rainforest and forest fringes. The Carpet Viper and Gaboon Viper are terrestrial while the Green Tree Viper is arboreal. Other species in this group includes the Puff adder *(Bitis arietans)* which is mostly found in the savanna regions and has a terrestrial and nocturnal habit as well as the Boomslang (*Dispholidus typus*) which is also found in savannah regions but is arboreal in nature. Species in this category have significant moderate to severe coagulating and hemorrhaging effects resulting in severe bleeding after bites and may be associated with severe blistering and pain at bite site (Table 4). It is observed that, there is availability of antivenoms for the species with haemoraghins except the West African Green Tree Viper for which there is no antivenom (Table 4). The Puff adder and the West African Tree Viper are WHO Cat 1 species and are highly lethal. The haemoraghins characterized from these species are mostly zinc metalloproteinases.

**3.2.4. Snake species with procoagulant effects in humans.** Snake species with procoagulant toxins identified in the study area included all the species which have haemorrhagins

**Table 3. Snake species with cardiotoxic effects in humans [23, 24].**

| Common name (s) | Habitat/habit | Species-specific toxin | Clinical manifestation | Antivenom availability |
|---|---|---|---|---|
| Variable Burrowing Asp (*Atractaspis irregularis*) * | Tropical and moist forest terrain, Savanna regions at the edge of forest and at elevations up to about 1800 m. Fossorial and nocturnal | Cardiotoxins (endothelin-like activity of sarafotoxin) Necrotoxins | Local pain, swelling, bruising, blistering and necrosis Ischaemia or infarction | No Antivenoms |
| Gaboon Viper (*Bitis rhinoceros*)** | Rainforest, secondary forest and forest fringes up to about 2000 metres elevation Terrestrial, often found in leaf litter where its colour and pattern provide excellent camouflage | Procoagulants (Fibrinogenases) Anticoagulants Haemorrhagins (Zinc metalloproteinase) Cardiotoxins (Present but not defined) Necrotoxins (Present but not defined) | Cardiotoxicity— coagulopathy haemorrhages 60–80% rate of envenoming Marked local effects; pain, severe swelling, bruising, blistering Minor neurotoxic paralysis | FAV-Afrique Antivipmyn Africa Polyvalent Snake Venom Antiserum SAIMR Polyvalent Antivenom SII Polyvalent Antisnake Venom Serum (lyophilized) |

[+ = present but not defined

* = WHO category 1—Highest medically important venomous snakes

** = WHO category 2—Secondary medically important highly venomous snakes].

**Table 4. Snake species with haemorrhagic (haematotoxic) manifestations in humans [23, 24].**

| Common name | Habitat/habit | Other toxins | Clinical manifestation | Antivenom availability |
|---|---|---|---|---|
| West African Carpet Viper (*Echis ocellatus*)** | Terrestrial, nocturnal and crepuscular | Anticoagulants, Nephrotoxins, Mixture of procoagulants, Necrotoxins, | Renal complication, secondary to coagulopathy, Marked local effects (pain, severe swelling, bruising, blistering), Moderate to severe necrosis | SAIMR Echis Antivenom, FAV-Afrique, Pan-African antivenom (EchiTAb-Plus-ICP), EchiTAbG, Antivipmyn Africa, SAIMR Echis antivenom |
| West African Green Tree Viper (*Atheris chloroechis*)* | Forest, semi-arid areas, Diurnal and arboreal | | Severe envenoming possible, potentially lethal, Local pain, swelling, bruising & blistering, Coagulopathy & hemorrhages are uncommon to rare, but may be moderate to severe coagulopathy | No Antivenoms |
| Puff adder (*Bitis arietans*)* | Savanna, nocturnal | Procoagulants Anticoagulants Nephrotoxins Necrotoxins+ | Marked local effects; pain, severe swelling, bruising, blistering, Moderate to severe local Necrosis, Common, moderate to severe coagulopathy + haemorrhagins causing extensive bleeding, Cardiotoxicity is common and is major clinical effect, Shock secondary to fluid shifts due to local tissue injury is likely in severe cases | SAIMR Snakebite Kit, Antivipmyn Africa, Pan-African antivenom (EchiTAb-Plus-ICP), Polyvalent Snake Venom Antiserum, FAV-Afrique, Favirept, SAIMR Polyvalent Antivenom |
| Gaboon Viper (*Bitis rhinoceros*)** | Forest, nocturnal | Procoagulants, Fibrinogenases) Anticoagulants Cardiotoxins+ Necrotoxins+ | Marked local effects; pain, severe swelling, bruising, blistering, Minor neurotoxic paralysis, Coagulopathy & Haemorrhages, very common, coagulopathy + haemorrhagins causing bleeding is major clinical effect, Cardiotoxicity common, | FAV-Afrique, Antivipmyn Africa, Polyvalent Snake Venom Antiserum SAIMR Polyvalent Antivenom SII Polyvalent |
| Boomslang (*Dispholidus typus*\*) | Open bush, savannah, wooded grassland, Diurnal and arboreal | Procoagulants Anticoagulants | Local pain, swelling & bruising Coagulopathy & Haemorrhages Common, moderate to severe coagulopathy + haemorrhagins causing extensive bleeding | SAIMR Boomslang Antivenom |

[+ = present but not defined

* = WHO category 1—Highest medically important venomous snakes

** = WHO category 2—Secondary medically important highly venomous snakes].

except the West African Green Tree Viper, (*Atheris chloroechis*). Thus, the West African Carpet Viper (*Echis ocellatus*), West African Gaboon Viper (*Bitis rhinoceros*), Puff adder *(Bitis arietans) and* the Boomslang all have both haemorrhagic and procoagulant toxins (Table 4). These species are mostly found in either savannah or rainforest and forest fringes and are terrestrial in nature except the boomslang *Dispholidus typus* which is arboreal (Fig 3). Species in this category belong to both WHO Cat 1 and 2 making them highly lethal (Table 4). Cardiotoxicity is widely associated with snake bites from these species due to potential synergistic effects of the procoagulant and hemorrhagic toxins with direct cardiotoxicity.

**3.2.5. Snake species with cytotoxic and broad-spectrum toxicity in humans.** Several snake species encountered in the study possess toxins with cytotoxic properties and a broad range of toxicity including nephrotoxicity and neurotoxicity. These include Rhinoceros Viper (*Bitis nasicornis*), West African Green Tree Viper, (*Atheris chloroechis*), West African Carpet Viper (*Echis ocellatus*), West African Gaboon Viper (*Bitis rhinoceros*), Puff adder *(Bitis arietans)*, Variable Burrowing Asp, Senegalese cobra and the Boomslang. Their clinical manifestation includes rapid and conspicuous swelling, intense pain, severe shock, and local blistering. The synergistic effects of these broad-acting toxins can result in the patient's blood becoming incoagulable leading to haematuria, haematemesis, uncoordinated movements, defecation, urination, swelling of the tongue and eyelids and convulsions (Table 5). This is particularly evident in the Rhinoceros Viper (*Bitis nasicornis*) a CAT 1 species which is a terrestrial and nocturnal snake.

**Table 5. Snake species with cytotoxic and broad-range toxicity in humans [23, 24].**

| Species | Habitat/habit | Species-specific toxin | Clinical manifestation | Antivenom availability |
|---|---|---|---|---|
| West African Carpet Viper (*Echis ocellatus*)** | Terrestrial, nocturnal & crepuscular | Mixture of procoagulants, Anticoagulants, Haemorrhagin (Zinc metalloproteinase), Nephrotoxins, Necrotoxins | Renal complication, coagulopathy, Marked local effects (pain, severe swelling, bruising, blistering), Moderate to severe necrosis | SAIMR Echis Antivenom, FAV-Afrique, Pan-African antivenom (EchiTAb-Plus-ICP), EchiTAbG, Antivipmyn Africa, SAIMR Echis antivenom |
| West African Green Tree Viper (*Atheris chloroechis*)* | Forest, semi-arid regions, Diurnal & arboreal | Haemorrhagins | Severe envenoming possible, potentially lethal<br>Local pain, swelling, bruising & blistering<br>Coagulopathy & hemorrhages are uncommon to rare, but may be moderate to severe coagulopathy | No Antivenoms |
| Variable Burrowing Asp (*Atractaspis irregularis*)* | Moist forest, Savanna, Fossorial & nocturnal | Cardiotoxins (Indirect cardiotoxin (endothelin-like activity of sarafotoxins), Necrotoxins | Local pain, swelling, bruising & blistering<br>Local Necrosis is common but usually not severe<br>Ischaemia or infarction | No Antivenoms |
| Puff adder (*Bitis arietans*)* | Savanna & scrubs | Procoagulants, Anticoagulants, Haemorrhagins+, Nephrotoxins, Necrotoxins+ | 60–80% rate of envenoming<br>Marked local effects; pain, severe swelling, bruising, blistering<br>Moderate to severe local Necrosis<br>Common, moderate to severe coagulopathy + haemorrhagins causing extensive bleeding<br>Cardiotoxicity is common and is major clinical effect<br>Shock secondary to fluid shifts due to local tissue injury is likely in severe cases | SAIMR Snakebite Kit, Antivipmyn Africa, Pan-African antivenom (EchiTAb-Plus-ICP), Polyvalent, FAV-Afrique, Favirept, SAIMR, Polyvalent Antivenom |
| Rhinoceros viper (*Bitis nasicornis*)** | Forest, woodland, nocturnal good swimmer | Cytotoxin | Rapid and conspicuous swelling, intense pain, severe shock, and local blistering. Other symptoms include uncoordinated movements, defecation, urination, swelling of the tongue and eyelids, convulsions, and unconsciousness. Blistering, bruising, and necrosis may be extensive. Sudden hypotension, heart damage, and dyspnoea may occur<br>The blood may become incoagulable, with internal bleeding that may lead to haematuria and haematemesis | SAIMR Snakebite Kit, Antivipmyn Africa, Pan-African antivenom (EchiTAb-Plus-ICP), Polyvalent Snake Venom Antiserum, SAIMR Polyvalent Antivenom |
| West African Gaboon Viper (*Bitis rhinoceros*)** | Forest and forest fringes | Procoagulants (Fibrinogenases), Anticoagulants, Haemorrhagins (Zinc metalloproteinase), Cardiotoxins+, Necrotoxins+ | Marked local effects; pain, severe swelling, bruising, blistering, Minor neurotoxic paralysis, Coagulopathy & Haemorrhages, very common, coagulopathy + haemorrhagins causing bleeding is major clinical effect, Cardiotoxicity, Common, cardiotoxicity is major clinical effect | FAV-Afrique, Antivipmyn Africa Polyvalent Snake Venom Antiserum SAIMR Polyvalent Antivenom SII Polyvalent |
| Boomslang (*Dispholidus typus*)** | Open bush, savanna and sparsely wooded grassland, Diurnal and arboreal | Procoagulants, Anticoagulants, Haemorrhagins | Local pain, swelling & bruising, Coagulopathy & Haemorrhages, Common, moderate to severe coagulopathy + haemorrhagins causing extensive bleeding | SAIMR Boomslang Antivenom |

**3.2.6: Snake species and myotoxicity in humans.** There is insufficient evidence associating sub-Saharan Africa and Ghanaian snake venoms with myotoxins. Available data presently ascribes this toxin to sea snakes including *Hydrophis curtus*, selected bitis spp., among others (Table 6).

**Table 6. Snake species and myotoxicity in humans[†].**

| Scientific name | Common name | Effect |
|---|---|---|
| **ELAPIDAE** | | |
| *Micropechis* **ikaheka** | New Guinea small eyed snake | Moderate systemic myolysis |
| *Micrurus* **spp. (selected)** | Selected South American coral snakes | Moderate systemic myolysis |
| *Notechis* **spp.** | Australian tiger snakes | Severe systemic myolysis |
| *Oxyuranus* **spp.** | Australian taipans | Moderate systemic myolysis |
| *Pseudechis* **spp.** | Australian mulga & black snakes | Moderate to severe systemic myolysis |
| *Tropidechis carinatus* | Rough scaled snake | Severe systemic myolysis |
| | Various Sea snakes | Moderate to severe systemic myolysis |
| **VIPERIDAE** | | |
| *Bothrops* **spp. (selected)** | Selected species of South American pit vipers | Moderate systemic myolysis |
| *Crotalus* **spp. (selected)** | Selected species of South American rattlesnakes | Moderate systemic myolysis |
| *Daboia russelii pulchella* | Sri Lankan Russell's viper | Moderate systemic myolysis |

†: Snakes known to have significant systemic myotoxins of medical significance. This list is however not exhaustive.

## 3.3: Anti-Venom availability

Our review identified 15 snake antivenom that have been used in the management of snakebite envenomation, 13 of which are in active production presently (Table 7). We noted that, FAV-Afrique, which is widely used and is touted as the most effective against some medically important snakes prevalent in Ghana and the sub-region, was discontinued by the manufacturer Sanofi Pasteur in 2014. Attempts to re-start production of this antivenom is yet to materialize [25].

In Ghana, three antivenoms are registered with the Food and Drug Authority of Ghana [26]. These are Snake Venom Antiserum (African) (AFRIVEN-10), Snake Venom Antiserum (Pan African) (PANAF-Premium) and the Anti-Snake Venom Serum Pan Africa [10] BEAFRIQUE-10. The AFRIVEN-10 and PANAF-Premium are lyophilized powders for reconstitution while BEAFRIQUE-10 is a liquid formulation. All three are polyvalent with PANAF-Premium being the most broadly-acting antivenom with antivenom activity against 14 species belonging to the *Bitis*, *Naja*, *Dendroaspis* and the *Echis* species. However, none of these three antisera is locally manufactured.

## 4 Discussion

Climate change is evoking increasing human snake encounters and hence a rise in snake bite incidence [29]. Many of these snakes are however, not venomous and may pass undetected or without signs of envenomation [30]. About a quarter of bites from venomous species do not necessarily result in systemic envenomation [4,16]. The few that may otherwise cause clinical signs and symptoms provide compelling reasons to study their distribution, habit, venom type and expression and clinical manifestation including the pathophysiology of venom action. This is the focus of this paper in guiding clinicians in Ghana and the West African sub-region in the management of snake bites and conservation scientists in protecting snakes from going extinct.

The clinical manifestations of snake bites envenomation albeit rare is broadly based on venom activity classified as neurotoxic, myotoxic, cardiotoxic, haemorrhagic, cytotoxic, procoagulant and nephrotoxic [2,15,31,32]. Envenomation may present with local or systemic clinical features according to the venom type.

**Table 7. Anti-snake venom sera availability [26–28].**

| S. No. | Type of antivenom | Manufacturer | Country of origin | Status of production currently | Snake species | Comment |
|---|---|---|---|---|---|---|
| 1 | FAV-Afrique | Sanofi Pasteur | France | Discontinued in January 2014. Last batch expired in Jan 2016 | 10 species of the Elapidae and Viperidae families | The most highly effective in treating envenoming by *Echis ocellatus*, the West African saw-scaled viper that causes great morbidity and mortality throughout the West and Central African savannah MicroPharm Limited acquired product in 2018 and seeking to re-launch it |
| 2 | SAIMR Polyvalent Antivenom | South African Vaccine Producers | South Africa | Active production | *B. arietans, B. gabonica, D. angusticeps, D. jamesoni, D. polylepis, H. haemachatus, N. annulifera, N. melanoleuca, N. mossambica, N. nivea* | Considered as the current "Gold Standard" for treating envenomings by many African snake species |
| 3 | Antivipmyn Africa | Instituto Bioclon / Silanes | Mexico | Active production | *B. arietans, B. gabonica, B. rhinoceros, D. angusticeps, D. jamesoni, D. polylepis, D. viridis, E. leucogaster, E. ocellatus, E. pyramidum, N. haje, N. katiensis, N. melanoleuca, N. nigricollis, N. nivea* | |
| 4 | Bivalent Naja / Walterinnesia Snake Antivenom | National Antivenom and Vaccine Production Centre | Saudi Arabia | Active production | *N. arabica* *N. haje* *N. melanoleuca* *N. naja* *N. nigricollis* *N. nivea* *Walterinnesia aegyptia* *Walterinnesia morgani* | |
| 5 | Pan-African antivenom (EchiTAb-Plus-ICP) | Instituto Clodomiro Picado (ICP) University of Costa Rica | Costa Rica | Active production | *B. arietans, E. ocellatus, N. nigricollis D. polylepis) N. mossambica N. annulifera H. haemachatus* | Intact IgG The expanded-scope EchiTAb + ICP has lower ability to neutralize the venom of *B. arietans*, but similar ability to neutralize the venoms of *D. polylepis, N. mossambica* and *H. haemachatus* When compared to FAV Afrique and the SVA African antivenoms |
| 6 | Snake Venom Antiserum | Bharat Serums & Vaccines | India | | *B. caeruleus N. haje* | |
| 7 | Inoserp Pan-Africa | Inosan Biopharma | Mexico | Active production | *E. ocellatus, E. leucogaster, E. pyramidum, B. arietans, B. rhinoceros, B. nasicornis, B. gabonica D. polylepis, D. viridis, D. angusticeps, D. jamesoni, N. nigricollis, N. melanoleuca, N. haje, N. pallida N. nubiae, N. katiensis N. senegalensis.* | Lyophilized Polyvalent F(ab')2 Immunoglobulin Fragments (Equine) |

*(Continued)*

**Table 7.** (Continued)

| S. No. | Type of antivenom | Manufacturer | Country of origin | Status of production currently | Snake species | Comment |
|---|---|---|---|---|---|---|
| 8 | Favirept Polyvalent Snake Antivenin | Sanofi Pasteur | France | Discontinued | B. arietans<br>C. cerastes<br>D. deserti<br>E. leucogaster<br>N. nigricollis,<br>N. melanoleuca, N. haje, | |
| 9 | SII Polyvalent Antisnake (Snake Antivenom Seum IP) | Serum Institute of India | India | Active production | N. naja<br>E. carinatus<br>D. russelii<br>B. caeruleus | Polyvalent equine antivenom |
| 10 | SAIMR Echis Antivenom | South African Vaccine Producers | South Africa | Active production | E. carinatus,<br>E. ocellatus,<br>E. coloratus, Cerastes spp. | |
| 11 | EchiTAbG | MicroPharm | United Kingdom | Active production | Echis ocellatus | |
| 12 | SAIMR Boomslang Antivenom | South African Vaccine Producers | South Africa | Active production | Dispholidus typus | |
| 13 | Snake Venom Antiserum (African) (AFRIVEN-10) | VINS Bioproducts | India | In active production | B. arietans,<br>B. gabonica,<br>E. ocellatus,<br>E. leucogaster, N. haje,<br>N. melanoleuca, N. nigricollis,<br>D. polylepis,<br>D. viridis,<br>D. jamesoni. | Lyophilized powder<br>Polyvalent, enzyme-refined equine antivenom immunoglobulin fragments |
| 14 | Snake Venom Antiserum (Pan African) (PANAF-Premium) | Premium Serums and Vaccines | India | In active production | B. arietans,<br>B. gabonica,<br>B. nasicornis,<br>B. rhinoceros<br>D. angusticeps, D. jamesoni,<br>D. polylepis,<br>D. viridis,<br>E. carinatus,<br>E. leucogaster, E. ocellatus,<br>N. nigricollis,<br>N. haje,<br>N. melanoleuca | Lyophilized powder<br>Polyvalent, enzyme-refined equine antivenom immunoglobulins |
| 15 | Anti-Snake Venom Serum Pan Africa [10] BEAFRIQUE-10 | M/s. Biological E | India | In active production | B. arietans,<br>B. gabonica,<br>E. ocellatus,<br>E. leucogaster, N. haje,<br>N. melanoleuca, N. nigricollis,<br>D. polylepis,<br>D. viridis,<br>D. jamesoni. | Liquid,<br>Polyvalent, Enzyme refined, Equine F(ab') immunoglobulin |

Snakes that express neurotoxic venoms are the most prevalent across Ghana though less represented in the Forest-Savanna Transition (FST) zone. Haemorrhagic toxin expressing snakes were just less prevalent than their neurotoxic toxin expressing counterparts but with a marked presence in the FST zone and not very much in the Moist and Wet Evergreen ecological areas of the country. The procoagulant toxin expressing snakes show similar distribution in the FST zone just as the "haemorrhagins" with similar or same species present. Cytotoxic

venom expressing species were quite confined to the moist and wet evergreen ecological zones. This distribution is shared by cardiotoxic venom expressing snakes but with additional and limited distribution to deciduous forest and guinea savannah ecological zones.

This distribution pattern may very well provide relevant epidemiological information for identification of snake bite and venom type and can guide clinical management. However, clinicians should interpret this distribution pattern with a lot of care and circumspection as the changing climate may cause a range shift in some of the species. Warmer climates are predicted to help snakes attain the optimum temperatures for digestion and reproduction because they are ectothermic and rely on their surrounding temperatures to provide heat [33]. Studies have found that longer periods of warmer temperatures could give rattle snakes a longer active season with more time to hunt and feed [34].

Envenomation from neurotoxic venom expressing snakes, typically elapids may show any of the following symptoms: vomiting, dizziness, blurred vision, weakness, inability to walk, tightness of the throat and tingling sensation of the body. Eye signs may include palpebral ptosis, ophthalmoplegia and facial muscle weakness. Delafontaine, captured eye signs in what they referred to as venom ophthalmia consisting of "pain, hyperemia, blepharitis, blepharospasm and corneal erosions" [17]. Neurotoxic venoms may also show local signs including numbness/ paresthesia and pain at the site of bite as well as local edema/swelling and erythema.

Bites from neurotoxin-expressing snakes may manifest systemically as muscle weakness, dysphagia, dyspnoea, generalized myalgia, possible neuromuscular paralysis and salivation. It is prudent for the health care team to appreciate the red flags for systemic neurotoxic envenomation including respiratory difficulty/paralysis, bulbar paralysis and respiratory failure. This may require intensive care facilities [35–41].

Occasionally, there are unusual presentations such as "transitory loss of taste" as reported by Bucaretchi, et al. [35] and "early morning neuroparalytic syndrome (EMNS)" as described by Anadure [42]. It is imperative for the health care team to look out for these uncommon presentations for prompt intervention where necessary.

Typically, elapidaes express neurotoxic venoms. However, these ophidia never cease to evolve surprises as it have been showed that the venom of the black-necked spitting cobra, *Naja nigricollis* demonstrated cardiotoxic effects [43,44]. In another twist, the following African spitting cobras, *Naja mossambica*, *N. nigricincta*, *N. nigricollis*, *N. pallid*, *N. ashei*, *N. katiensis*, *and N. nubiae* demonstrated potent anticoagulant activity in their venom arising from strong inhibition of Factor Xa [45,46]. Unfortunately, it was realized that the anticoagulatory activity of these elapids is not particularly neutralized by current antivenoms and this may pose problems in managing envenomation [45]. Additional work by others have revealed that the venoms of *Naja nigricollis* and *Naja ashei*, both African spitting cobras demonstrated cytotoxic activity as well [47, 48]. Thus, the care team must be thorough in their history, examination and investigations as well as reconciling the epidemiology and clinical presentations in patients to arrive at the most likely cause of the envenomation and hence appropriate management with some sense of the complications to expect.

Some of these atypical presentations of selected elapids mimic venom activities of ophidians in the family Viperidae. This family is principally made up of vipers of varying species and the West African carpet viper (*Echis ocellat*us) is reported to cause more deaths than any other snake in sub-Saharan Africa [22,49]. Envenomation from viper bites may result in local and systemic reactions including local pain, swelling/oedema, bruising, blistering, necrosis, haemorrhage, coagulopathy and nephrotoxicity as well as cardiovascular disturbances like hypotension as a sequelae. Cardiotoxicity is a rare complication but does occur [50,51].

Agarwal, A., et al. [19] reported "sinus node dysfunction" complicating a viper bite. This cardiotoxic manifestation of the venom showed sinus arrest and junctional escape rhythm on the ECG. This cardiotoxic manifestation of viper venom was corroborated by Chara, et al [52] in a case of acute myocardial infarction and/or ischaemia and myocarditis in a patient following a viper bite. A plethora of events including acute heart failure with acute pulmonary oedema, a state of cardiogenic shock, accompanied by multi-organ failure, intravascular disseminated coagulation and neurological damage were recognised. It is thus imperative for clinicians to include in their search for causes of abnormal cardiac events in patients, envenomation from viper bites.

Typically, vipers also express venoms with haemorrhagic manifestations in humans. Clinical expressions may be mediated through coagulopathic toxins (procoagulant, anticoagulant), nephrotoxins and necrotoxins [19, 32, 53–55]. Expressed venoms with haemorrhagic effects may present with local and/or systemic symptoms and signs.

Local presentations may include but not limited to the following: local pain, swelling/oedema, bruising, blistering, local necrosis, tissue destruction, local bleeding and subcutaneous bleed [18, 22,44, 55–57].

Usually, the systemic manifestations of expressed venoms with haemorrhagic effects result from their coagulopathic and haemorrhagic properties. Systemic presentations are very varied, affecting nearly every organ and organ system of the body. The following have been catalogued as some of the systemic manifestations of venoms with haemorrhagic effects: Bilateral pulmonary embolism, venom induced consumption coagulopathy (VICC) with intracranial haemorrhage (ICH) [22,58], brainstem ischemic stroke [57], acute kidney injury (AKI) and anuric renal failure [55,59], encephalopathy [59], internal bleeding [22], thoracic pain, respiratory failure, systemic bleeding syndrome, haemorrhagic shock, and cardiovascular collapse [22,59–61]. Meanwhile, Cao and colleagues also reported intestinal necrosis following a North American pit viper bite while haematemesis and hypovolaemic shock has also been documented [21]. There is also the rather curious twist of bilateral hyphema and angle closure attack that has been reported [62].

Typically, venom from the snake family Viperidae are known for their coagulopathic and haemorrhagic properties at the neglect of other potentially harmful sequelae of viper bites. Youngman, et al [63,64] demonstrated strong neurotoxic effects with the venom of the Bitis genus of snakes including *B. atropos*, *B. caudalis*, *B. arietans*, *B. armata*, *B. cornuta*, *B. peringueyi and B. rubida*. This effect was largely mediated through phospholipase $A_2$ (PLA$_2$). Myotoxicity was also evident in some Bitis species.

This list of local and systemic features of venoms with haemorrhagic manifestations is not exhaustive and clinicians may very well acquaint themselves with the multi-organ effects of these venoms for prompt recognition of envenomation and appropriate management. Clinicians must also display a high sense of alertness for atypical presentations.

Snakes expressing venoms with cytotoxic and broad-range toxicity quite well overlap with ophidias expressing venoms with haemorrhagic manifestations but for Rhinoceros viper and the Variable Burrowing Asp (*Atractaspis irregularis*). The later doubles as a cardiotoxin venom expressing snake. In effect, the local and systemic clinical features of envenomation are quite similar to those earlier described in the aforementioned categories. The Rhinoceros viper and the Variable Burrowing Asp (*Atractaspis irregularis*) express their clinical features through species specific toxins including cytotoxins and cardiotoxins, necrotoxins and sarafotoxins respectively. The venom of the Rhinoceros viper is rather ferocious causing rapid and conspicuous swelling), intense pain, local blistering, bruising, and necrosis that may be extensive. Some documented systemic features include the following: swelling of the tongue and eyelids, defecation, urination, uncoordinated movements, heart damage, sudden hypotension, severe

shock) with rapid deterioration of consciousness. Dyspnoea may occur. There has been reports of coagulopathies with internal bleeding, haematuria and haematemesis [1,23].

There is insufficient evidence associating sub-Saharan Africa and Ghanaian snake venoms with myotoxic venoms. However, this should be applied with circumspection as the changing climate may cause a range shift in some of the species. Global warming, globalisation and the flourishing snake pet markets make the threat of non-native snakes in West Africa and Ghana even more compelling. Clinicians must therefore have an idea of the clinical manifestations of myotoxic snake venoms including features of rhabdomyolysis and muscle necrosis [18,20]

Our study identified the clinical features of envenomation, snake type and venom expressed. This will help the attending clinician in making decisions on treatment of SBE. However, this manuscript discussed only those snake antivenoms that are available on the market and some pearls in their use. The full complement of treating snake bites is beyond the scope of this paper. In considering the use of snake antivenom, choice should be based on clinical presentation and evolution of symptoms rather than on snake identification alone [22]. Clinicians should also be mindful of reports speaking to the limited activity of antivenoms on local effects of the toxins in contrast to their neutralising effects on systemic manifestations [65,66].

It is worthy to note that most of the available snake antivenoms identified in this review are polyvalent in nature and are marketed to be active agents of a variety of snake species. Indeed, the Inoserp Pan-Africa anti-venom is marketed to be effective against envenomation of 14 different snake species. These polyvalent antivenom very well suit the African region which has a variety of snake species causing envenomation in different agro-ecological zones. Bearing in mind the economic challenges in ensuring adequate stocking of pharmaceuticals in these regions, such polyvalent antivenoms provide one-stop options for managing several envenomation in contrast to having to stock specific antivenom for each species of medically important snakes found in the agroecological zones. That challenge however, will be ensuring that these snake antivenom meet the required quality standards especially for liquid formulated antisera which requires strict maintenance of the cold-chain throughout storage period; a situation that can be challenging in limited resource settings. It would therefore require effective post marketing surveillance to provide the needed quality assurance while also ensuring that the cold-chain is maintained throughout the shelf life of these liquid snake antivenom. Furthermore, regular updates on routine quality assessment of these antisera should be made public by regulatory agencies.

In Ghana 3 polyvalent antivenoms were cited according to records of the Food and Drug Authority of Ghana of which the Snake Venom Antiserum (Pan African) (PANAF-Premium) has the broadest coverage of antivenom activity spanning 14 different species in the Viperidae and Elapidae families. The fact that these antivenoms are lyophilized powders for reconstitution inures to the benefit of the snakebite envenomation in the region by ensuring availability because of difficulties in maintaining the cold-chain liquid antivenom formulations require to remain efficacious. Also of important note is the fact that there are no antivenom manufacturers in Ghana and this poses the risk of unavailability of supply in the event of global logistic challenges as was seen with the recent COVID-19 pandemic and halting of production of antivenoms by some manufacturers. An assured supply especially to the hotspots will certainly optimise management of SBE in the country.

## 5 Conclusion

There is usually insufficient documentation of snake bite in health records in Ghana. This certainly impacts on the management of snake bite envenoming (SBE). Thus, an understanding

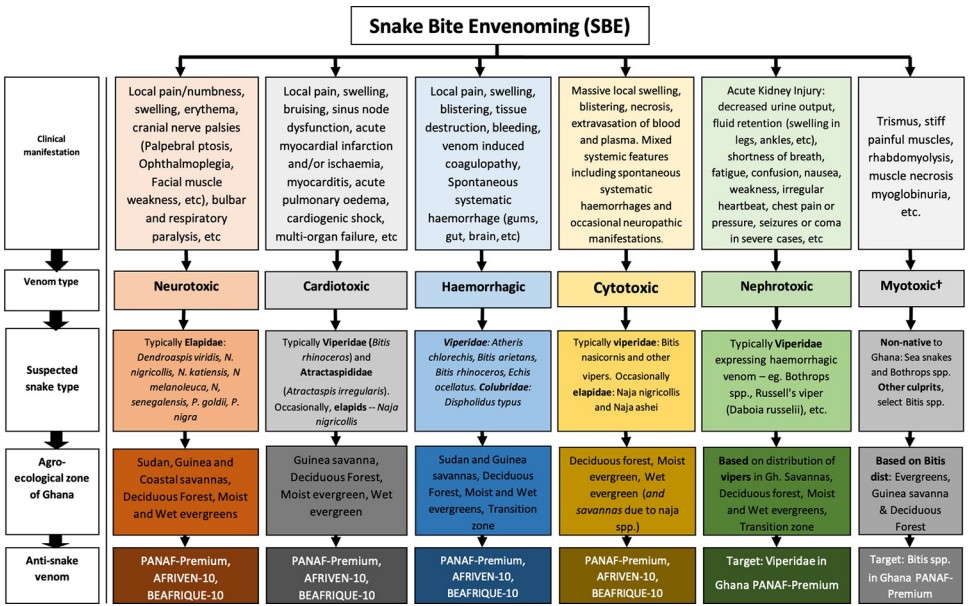

**Fig 4. SBE Algorithm † Gradually increasing selected species identified to manifest myotoxic features.**

of the maze of expressed venoms including cardiotoxic, neurotoxic, myotoxic, nephrotoxic, and haemotoxic and their clinical manifestation as well as the epidemiology of medically important snake species in the country will in no doubt improve diagnoses, early and appropriate treatment of SBE. Careful interpretation of clinical symptoms and signs including atypical presentations will afford better care of patients with SBE. This is demonstrated in Fig 4.

A patient presenting in a health facility typically shows clinical features mimicking one of several groups of SBE. The manifestation may be mixed, quite often for local signs but will certainly display selected clinical features consistent with a particular type of expressed venom. This together with or without a description of the offending snake as well as the agro-ecological zone that the incident occurred may give of the likely snake type and expressed venom. This will provide the type of management to pursue including observation for non-venomous bites, treatment of local signs that are usually not very responsive to systemic anti-venom and the administration of appropriate anti-snake sera for systemic manifestation of envenoming. This study provides a useful guide in the management of SBE and thus reducing morbidity, disability and mortality due to snake bites in the country and the West African sub-region.

Meanwhile, the continuing availability of antivenom, which is the principal treatment for SBE remains questionable in Ghana and the sub-region. South Africa remains the only country on the African continent actively producing snake bite antivenom. It will be important to build the capacity of African pharmaceutical companies to be able to produce these antivenoms in order to safeguard the supply as a long-term strategy of dealing with the neglected snake bite envenomation.

## 6 Limitations of study

Data from the District Health Information Management System (DHIMS) of the Ghana Health Service (GHS), which was used in this study exists as summarised information thus lack specific characteristics of snakes and snake bites in Ghana. Furthermore, in applying the conclusions from this current study circumspection must be exercised in view of the impact of climate change on snake distribution and atypical presentations of some snake venoms.

## 7 Recommendations

This work revealed interesting characteristics of snake venoms. It also exposed some gaps that need further studies for in-depth understanding especially among native snakes in Ghana and the West African sub-region. Among these are studies into the atypical presentations of Elapidae and Viperidae including further characterisation of their venoms. The nephrotoxic activity of selected species of Viperidae native to Ghana and the sub-region also need some further studies to enhance clinical care of patients presenting to health facilities with renal manifestations following snake bite. Snakes with expressed venoms demonstrating myotoxic activities are on record to be largely non-native to Ghana and the sub-region. Some literatures have however, shown venom from selected Bitis species to exhibit myotoxic activity and this needs further studies. Subsequent manuscripts should address some of these gaps.

## Acknowledgments

We acknowledge Mrs. Mary Ayiah-Adjei and many colleagues for their time during literature search and data collection and also for providing feedback on earlier drafts. We appreciate the support from the Ghana Health Service (GHS) in providing advise and permitting us to use information on their database for this study.

## Author Contributions

**Conceptualization:** Justus Precious Deikumah, Robert Peter Biney, Mawuli Kotope Gyakobo.

**Data curation:** Justus Precious Deikumah, Robert Peter Biney, John Koku Awoonor-Williams, Mawuli Kotope Gyakobo.

**Formal analysis:** Justus Precious Deikumah, Robert Peter Biney, Mawuli Kotope Gyakobo.

**Methodology:** Justus Precious Deikumah, Robert Peter Biney, John Koku Awoonor-Williams, Mawuli Kotope Gyakobo.

**Project administration:** Justus Precious Deikumah, Robert Peter Biney, Mawuli Kotope Gyakobo.

**Writing – original draft:** Justus Precious Deikumah, Robert Peter Biney, Mawuli Kotope Gyakobo.

**Writing – review & editing:** Justus Precious Deikumah, Robert Peter Biney, John Koku Awoonor-Williams, Mawuli Kotope Gyakobo.

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
