## [Decision Letter · Decision Letter 0]

27 Jan 2023

Dear Gyakobo,

Thank you very much for submitting your manuscript "Compendium of medically important snakes, venom activity and clinical presentations in Ghana" for consideration at PLOS Neglected Tropical Diseases. As with all papers reviewed by the journal, your manuscript was reviewed by members of the editorial board and by several independent reviewers. In light of the reviews (below this email), we would like to invite the resubmission of a significantly-revised version that takes into account the reviewers' comments. 

We cannot make any decision about publication until we have seen the revised manuscript and your response to the reviewers' comments. Your revised manuscript is also likely to be sent to reviewers for further evaluation.

Sincerely,

Wuelton M. Monteiro, Ph.D.

Section Editor

Wuelton Monteiro

Section Editor

Reviewer's Responses to Questions

**Key Review Criteria Required for Acceptance?**

**Methods**

-Are the objectives of the study clearly articulated with a clear testable hypothesis stated?

-Is the study design appropriate to address the stated objectives?

-Is the population clearly described and appropriate for the hypothesis being tested?

-Is the sample size sufficient to ensure adequate power to address the hypothesis being tested?

-Were correct statistical analysis used to support conclusions?

-Are there concerns about ethical or regulatory requirements being met?

Reviewer #1: (No Response)

Reviewer #2: 1. There is need to clarify how the information from literature databases was reviewed. Was it systematically reviewed? If so, it would be advisable to follow the PRISMA guidelines.

2. It would be helpful to provide more information on the sources of the data and how it was collected. For example, it is not clear how the data from the DHIMS database was augmented or what types of medical records were reviewed

3. It would be helpful to provide more information on the sampling method used for the study, including the criteria for selecting the personnel to be interviewed, the hospitals and health facilities to be included.

4. The study would need ethical clearance from an IRB

Reviewer #3: Please see the general comments.

**Results**

-Does the analysis presented match the analysis plan?

-Are the results clearly and completely presented?

-Are the figures (Tables, Images) of sufficient quality for clarity?

Reviewer #1: (No Response)

Reviewer #2: The results are presented clearly and are complete. The figures and tables are of sufficient quality for clarity. The analysis presented appears to match the analysis plan. However this may change after the review of the methodology. 

Some other general comments here include

• This sentence is not clear "Whereas a cardiotoxin, sarafotoxin, with endothelin-like properties has been isolated from the Variable Burrowing Asp"

• "Bites from these species present as coagulopathy and lead to infarction and ischemia." - It might be clearer to say "Bites from these species can cause coagulopathy, infarction, and ischemia

Reviewer #3: Please see the general comments.

**Conclusions**

-Are the conclusions supported by the data presented?

-Are the limitations of analysis clearly described?

-Do the authors discuss how these data can be helpful to advance our understanding of the topic under study?

-Is public health relevance addressed?

Reviewer #1: (No Response)

Reviewer #2: The conclusions is too broad and focus the main findings or implications of the study more. The authors need to be more concise and with the discussion points. The limitations of the analysis and study have not been included in this paper.

Reviewer #3: Please see the general comments.

**Editorial and Data Presentation Modifications?**

Reviewer #1: (No Response)

Reviewer #2: The overall paper may also need minor grammar checks

Reviewer #3: The manuscript ‘Compendium of medically important snakes, venom activity and clinical presentations in Ghana’ by Justus P. Deikumah and group is fairly an elaborate study and deserves publication in PNTD. The results of this study although do not present any new findings, however, give the compilation of various parameters. This study, I believe gives better confidence in the treatment and management of deadly snakebites in Ghana. The manuscript is well written, however with some minor grammatical errors and needs thorough correction before it is accepted.

**Summary and General Comments**

Reviewer #1: (No Response)

Reviewer #2: Introduction

Based on the review of relevant literature and data from the District Health Information Management System (DHIMS) of the Ghana Health Service (GHS), this study aims to provide an understanding of snake bite envenoming (SBE) in Ghana and the West African sub-region, including the distribution and habits of snake species, the types of venom they express, and their clinical manifestations. The study found that SBE in Ghana is largely grouped by the activity of the expressed venom, which can be neurotoxic, cardiotoxic, haemorrhagic, cytotoxic, myotoxic, nephrotoxic, or procoagulant. Neurotoxic venom is primarily expressed by elapid snakes, while viperidae snakes predominantly express venoms with cardiotoxic, haemorrhagic, nephrotoxic, and procoagulant activities. Venom expressing myotoxic activity is largely absent in the region, while venoms expressing cytotoxic activity are expressed by a variety of snake species, particularly viperidae. Snakes expressing neurotoxic and haemorrhagic venoms are prevalent across all agro-ecological zones in Ghana. The availability of antivenoms for SBE in the region is limited, with South Africa being the only African country actively producing antivenoms. Understanding the characteristics of snakes and their venoms in the region will be useful for the management of SBE and improving patient outcomes.

Strengths and weaknesses of the paper

Based on the information provided, it appears that the manuscript is a review of snake bite envenoming in Ghana, including information on snake distribution, venom expression, and clinical manifestations. The study used a desk review of published literature and data from the District Health Information Management System to gather information.

One strength of the manuscript is the inclusion of data from multiple sources, including published literature and a government health database. The review is also specific to the context of Ghana, which is useful for understanding the local epidemiology of snake bite envenoming.

One potential weakness of the manuscript is that it is based on a desk review rather than original research. It is also not clear whether the sample size for the data from the District Health Information Management System is sufficient to accurately represent the incidence of snake bite envenoming in Ghana. The methodology is generally not very clear.

Overall, the manuscript appears to provide useful information on snake bite envenoming in Ghana and could be valuable for clinicians and researchers working in the region. However, further original research may be necessary to fully understand the epidemiology of snake bite envenoming in Ghana. 

Additional comments for the authors

Here are other specific issues 

Introduction section:

• The study objectives is not clearly stated in the introduction. The lines 114 to 116 should be clearer and connect with materiel in the previous paragraph.

In the methods section:

• The criteria for selecting hospitals and regional medical stores is not described.

• The interviews with the experts will also need a standardised questionnaire.

• In the paragraph describing the data management process, it is not clear what is meant by "the matrix." It might be helpful to provide more context or explanation for this.

• It would be helpful to provide more information on the sources of the data and how it was collected. For example, it is not clear how the data from the DHIMS database was augmented or what types of medical records were reviewed.

• It would be helpful to provide more information on the sampling method used for the study, including the criteria for selecting the personnel to be interviewed, the hospitals and health facilities to be included.

• The interview with the experts will also need ethical clearance form an institutional review board.

• It might be helpful to provide more information on the study's limitations and any potential sources of bias or confounding factors ( overall limitations of the study and details can also be written after the discussion).

In the results section:

• It would be important to know what aspects the experts interviewed contributed to the results.

In the discussion section:

• The link between climate change and human-snake encounters is not well-established and seems like it has been misplaced here. It would be appropriate if the first paragraph focuses on the study findings.

• Generally, the discussion could be more concise and focused on the main findings and implications of the study.

• There is some instances of repetition of information from the results section which can be trimmed to make the discussion more focused.

• It will also be important to highlight what aspects where provided by experts and how they fit in the discussion.

• A section focusing on the limitations of the research, methodology should be written

In the conclusion section:

• The conclusion is too broad and does not clearly summarize the main findings or implications of the study.

• A separate section focusing on recommendations can be written

• The recommendation for building capacity for African pharmaceutical companies to produce antivenoms is not well-supported or explained.

for the major revisions;

 1. for the literature search - design a correct systematic review and report using PRISMA guidelines

 2. for the expert survey - inclusion and exclusion criteria, sample size determination, standardised questionnaire

Reviewer #3: The manuscript ‘Compendium of medically important snakes, venom activity and clinical presentations in Ghana’ by Justus P. Deikumah and group is fairly an elaborate study and deserves publication in PNTD. The results of this study although do not present any new findings, however, give the compilation of various parameters. This study, I believe gives better confidence in the treatment and management of deadly snakebites in Ghana. The manuscript is well written, however with some minor grammatical errors and needs thorough correction before it is accepted.

Minor comments

Line 154: 2.2. Methods, it is a repetition of line 124 (2. Methods)

Line 212; 3.2.1. Snakes species and neurotoxin in humans

This is misleading, change as neurotoxic symptoms in human

238 3.2.2. Snake species and Cardiotoxins in humans.

Change as cardiotoxicity or cardiotoxic effect in human

Table 3.2.1: Snakes species and neurotoxin (Postsynaptic) effects in humans

Change as neurotoxic

238 3.2.2. Snake species and Cardiotoxins in humans

Change as cardiotoxic effect

251 Table 3.2.2: Snakes species and Cardiotoxin effects in humans

Change as cardiotoxic

256 3.2.3. Snake species and haemorrhagins (or haematotoxins) in humans

Change as hemorrhagic effect

272 3.2.4. Snake species and procoagulants in humans

Change as procoagulant effect

289 3.2.5. Snake species and cytotoxic and broad-spectrum toxins in humans

Change as cytotoxic effect and broad-spectrum toxicity.

Discussion

Line 349, Snakes that express neurotoxins are the most prevalent across Ghana though 

Change this as neurotoxic venom.

Similarly change other properties all throughout the text.

Viz. cardiotoxicity or cardiotoxic effect (do not present as cardiotoxin or myotoxin or neurotoxin)

Lines 91 and 92 : Compounding these challenges is the proliferation of poor quality antivenoms available in these areas.

The current discussion justifying the above statement is inadequate, therefore throw little more light on this statement in the discussion. 

Important issue: the authors have given elaborate details on snakebite incidence and hot spots, clinical characteristics of snake venoms, the distribution pattern of medically important snakes of Ghana, snake species that induce various toxic properties such as neurotoxicity, myotoxicity, etc., available anti-venoms/anti-serum, however, the manuscript is seriously lacking about the information on mortality and morbidity details. Having acquired detailed data on the above-mentioned parameters, I believe, adding the mortality and morbidity details would elevate the quality of the manuscript. I suggest the authors, if possible, seriously consider the inclusion of the suggestion.

PLOS authors have the option to publish the peer review history of their article (what does this mean?). If published, this will include your full peer review and any attached files.

Reviewer #1: Yes: Pedro Ferreira Bisneto

Reviewer #2: No

Reviewer #3: Yes: Dr. K.Kemparaju

Professor

Department of Studies in Biochemistry

University of Mysore

Mysuru 570 006, India.
---

## [Decision Letter · Decision Letter 1]

23 Jun 2023

Dear Gyakobo,

We are pleased to inform you that your manuscript 'Compendium of medically important snakes, venom activity and clinical presentations in Ghana' has been provisionally accepted for publication in PLOS Neglected Tropical Diseases.

Best regards,

Wuelton M. Monteiro, Ph.D.

Section Editor

Wuelton Monteiro

Section Editor

Reviewer's Responses to Questions

**Key Review Criteria Required for Acceptance?**

**Methods**

-Are the objectives of the study clearly articulated with a clear testable hypothesis stated?

-Is the study design appropriate to address the stated objectives?

-Is the population clearly described and appropriate for the hypothesis being tested?

-Is the sample size sufficient to ensure adequate power to address the hypothesis being tested?

-Were correct statistical analysis used to support conclusions?

-Are there concerns about ethical or regulatory requirements being met?

Reviewer #1: (No Response)

Reviewer #2: comments in the general section

**Results**

-Does the analysis presented match the analysis plan?

-Are the results clearly and completely presented?

-Are the figures (Tables, Images) of sufficient quality for clarity?

Reviewer #1: (No Response)

Reviewer #2: comments in the general section

**Conclusions**

-Are the conclusions supported by the data presented?

-Are the limitations of analysis clearly described?

-Do the authors discuss how these data can be helpful to advance our understanding of the topic under study?

-Is public health relevance addressed?

Reviewer #1: (No Response)

Reviewer #2: comments in the general section

**Editorial and Data Presentation Modifications?**

Reviewer #1: (No Response)

Reviewer #2: (No Response)

**Summary and General Comments**

Reviewer #1: After reading the manuscript again, I can see that the authors improved it and adressed my suggestions. Therefore I can recommend its publication.

Reviewer #2: The authors have significantly improved the manuscript in line with the suggestions provided during the initial review phase. Their revisions have provided greater clarity to the content, enhancing the overall readability and scientific accuracy of the paper.

The previous concerns raised about the several aspects of the study have all been successfully addressed. The authors have expanded on key areas such as limitations and recommendations. This revised version of the manuscript provides a valuable and timely addition to the existing body of literature in snakebite envenomation , particularly for its relevance to the specific context of Ghana. The data presented here can serve as a robust reference for clinicians, researchers, and conservationists.

The tables and figures are well-structured and present essential information effectively. The manuscript is now more logically organized, and the argumentation flows well. Regarding language and grammar, it is apparent that the authors have put forth effort to enhance the text's clarity and precision. The manuscript is well-written, enabling readers from a variety of backgrounds to understand the implications of the study.

Overall, the authors have addressed the previously raised issues efficiently and effectively, substantially improving the quality and impact of their work. I am satisfied with the revisions and recommend that this paper be accepted for publication.

PLOS authors have the option to publish the peer review history of their article (what does this mean?). If published, this will include your full peer review and any attached files.

Reviewer #1: **Yes: **Pedro Ferreira Bisneto

Reviewer #2: No

---

## [Editor Report · Acceptance letter]

25 Jul 2023

Dear Gyakobo,

We are delighted to inform you that your manuscript, "Compendium of medically important snakes, venom activity and clinical presentations in Ghana," has been formally accepted for publication in PLOS Neglected Tropical Diseases.

Best regards,

Shaden Kamhawi

co-Editor-in-Chief

Paul Brindley

co-Editor-in-Chief
